# Study on Long-Term Tracing of Fibroblasts on Three-Dimensional Tissue Engineering Scaffolds Based on Graphene Quantum Dots

**DOI:** 10.3390/ijms231911040

**Published:** 2022-09-20

**Authors:** Tian Hou, Haiyang Ma, Xiang Gao, Haoyu Sun, Li Wang, Meiwen An

**Affiliations:** College of Biomedical Engineering, Taiyuan University of Technology, Taiyuan 030024, China

**Keywords:** graphene quantum dots, human skin fibroblasts, biocompatibility, fluorescence tracing, fluorescence attenuation, tissue engineering

## Abstract

In order to find a convenient and stable way to trace human skin fibroblasts (HSFs) in three-dimensional tissue engineering scaffolds for a long time, in this experiment, Graphene Oxide Quantum Dots (GOQDs), Amino Graphene Quantum Dots (AGQDs) and Carboxyl Graphene Quantum Dots (CGQDs) were used as the material source for labeling HSFs. Exploring the possibility of using it as a long-term tracer of HSFs in three-dimensional tissue engineering scaffolds, the contents of the experiment are as follows: the HSFs were cultured in a cell-culture medium composed of three kinds of Graphene Quantum Dots for 24 h, respectively; (1) using Cell Counting Kit 8 (CCK8), Transwell migration chamber and Phalloidin-iFlior 488 to detect the effect of Graphene Quantum Dots on the biocompatibility of HSFs; (2) using a living cell workstation to detect the fluorescence labeling results of three kinds of Graphene Quantum Dots on HSFs, and testing the fluorescence attenuation of HSFs for 7 days; (3) the HSFs labeled with Graphene Quantum Dots were inoculated on the three-dimensional chitosan demethylcellulose sodium scaffold, and the living cell workstation was used to detect the spatial distribution of the HSFs on the three-dimensional scaffold through the fluorescence properties of the HSFs.. Experimental results: (1) the results of CCK8, Transwell migration, and FITC-Phalloidin cytoskeleton test showed that the three kinds of Graphene Quantum Dots had no effect on the biological properties of HSFs (*p* < 0.05); (2) the results of the fluorescence labeling experiment showed that only AGQDs could make HSFs fluorescent, and cells showed orange–red fluorescence; (3) the results of long-range tracing of HSFs which were labeled by with AGQDs showed that the fluorescence life of the HSFs were as long as 7 days; (4) The spatial distribution of HSFs can be detected on the three-dimensional scaffold based on their fluorescence properties, and the detection time can be up to 7 days.

## 1. Introduction

The present situation of modern research on the repair of skin defects shows that the repair and reconstruction of skin tissue is closely related to the biological activity of HSFs, which is a dynamic and continuous process from cell to tissue [1,2]. The HSF as an important cell in the process of skin repair [3,4,5], combined with tissue-engineering technology, provides an effective scheme for the clinical treatment of skin defects and wounds [6,7]. The study of HSFs in tissue engineering scaffolds by cell-labeling technology is not only helpful to further clarify the mechanism of action between HSFs and tissue-engineering scaffolds, but also helps to optimize the conditions of HSFs implantation. By optimizing the method of evaluating the distribution, migration and differentiation of HSFs in tissue engineering, and tracing the living HSFs conveniently, effectively and stably, it will provide a better means to study the dynamic behavior of HSFs in tissue engineering scaffolds and provide a theoretical basis for the optimization of tissue engineering in clinical treatment of skin defects.

Because of their small size and close interaction with cells and intracellular components, luminescent nanomaterials are widely used in imaging applications in the fields of biology and medicine [8]. Among many nano-luminescent materials, Graphene Quantum Dots have been used in biological imaging because of their relatively small size, good water solubility, low biotoxicity and excellent luminescent properties [9]. Based on the purpose of the experiment and the excellent performance of Graphene Quantum Dots in biological imaging, GOQDs, AGQDs and CGQDs were selected as the source of cell fluorescence labeling materials. The HSFs were co-cultured with a gradient concentration of three kinds of Graphene Quantum Dots. The Graphene Quantum Dots which can be used as markers of HSFs were screened, the fluorescence attenuation of the labeled HSFs was detected and the fluorescence labeling time was obtained. The HSFs labeled with Graphene Quantum Dots were seeded on three-dimensional-tissue-engineering scaffolds to verify the possibility of detecting HSFs by fluorescence in a three-dimensional environment.

## 2. Results

### 2.1. Characterization of Three Kinds of Graphene Quantum Dots

#### 2.1.1. Detection of Graphene Quantum Dots by SEM and EDS

Three kinds of Graphene Quantum Dots were detected by the transmission electron microscope (TEM) and energy dispersive spectrometer (EDS). The results are shown in Figure 1. After the powders of CGQDs and GOQDs are dispersed in aqueous solution, they have a similar crystal morphology, such as the small black spots in the figure. The crystal shape of AGQDs in aqueous solution is a long rod, and its crystal shape is obviously different from the former two. It can be seen from Table 1 that the AGQDs is the only graphene quantum dot containing C, N and O at the same time. C, N and O are also the main elements of cell membrane.

#### 2.1.2. Results of Graphene Quantum Dots UV-Vis Spectrophotometer

Three kinds of Graphene Quantum Dots were detected by UV-vis spectrophotometer. As shown in Figure 2, the absorption wavelength of GOQDs is near 400 nm, and the absorption wavelength of AGQDs and CGQDs is near 450 nm.

### 2.2. Biocompatibility of Graphene Quantum Dots

#### 2.2.1. Results of CCK8 Test of Graphene Quantum Dots

The CCK8 was used to detect the activity of HSFs cultured for 24 h with three kinds of Graphene Quantum Dots, as shown in Figure 3. From the results of data decomposition, it was found that when the concentrations of the three Graphene Quantum Dots were 50 μg/mL, 100 μg/mL, 200 μg/mL and 400 μg/mL, there was no significant difference in fibroblast activity compared with the control group (*p* > 0.05).

#### 2.2.2. Detection of the Effect of Graphene Quantum Dots on Cytoskeleton

The cytoskeleton staining results of HSFs labeled with three kinds of Graphene Quantum Dots with gradient concentrations of 0 μg/mL (control group), 50 μg/mL, 100 μg/mL, 200 μg/mL and 400 μg/mL were shown in Figure 4, Figure 5 and Figure 6. The green fluorescent part (the filamentous structure in the picture) is the cytoskeleton, and the blue part is the nucleus. The results showed that the arrangement and trend of cytoskeleton showed the same state in the experimental group and the control group. The cytoskeleton is surrounded orderly by the nucleus, and its direction is parallel to the long axis of the cell. The cells in both the experimental group and the control group showed a long fusiform spread. The results showed that after HSFs were cultured in four concentrations of three kinds of Graphene Quantum Dots, the cytoskeleton arrangement and cell-spreading morphology were not affected. These three kinds of Graphene Quantum Dots have no toxic effect on the fibroblast cytoskeleton.

#### 2.2.3. Effect of Graphene Quantum Dots on Cell Migration

The Transwell migration chamber was used to detect the effects of three kinds of Graphene Quantum Dots on the migration ability of HSFs. The experimental results are as follows. Figure 7, Figure 8 and Figure 9 are the Transwell cell migration diagrams of the effects of three kinds of Graphene Quantum Dots on HSFs. Figure 10 shows the statistical chart of the number of HSFs migrating in the Transwell chamber. Through the statistics of the number of HSFs passing through the chamber, the results showed that there was no significant difference in the effect of the three Graphene Quantum Dots on the migration ability of HSFs (*p* > 0.05).

#### 2.2.4. Results of Fluorescence Labeling of Cells by Graphene Quantum Dots

In the experiment, HSFs were cultured in three kinds of Graphene Quantum Dots and five concentration gradients for 24 h, respectively, and the fluorescence properties of HSFs were detected by a living cell workstation. The results showed that only AGQDs could make HSFs labeled with fluorescence, and the fluorescence was orange–red, as shown in Figure 11. Fluorescence is mainly concentrated in the cytoplasm, and the nucleus does not have fluorescence.

#### 2.2.5. The Results of Fluorescence Attenuation of HSFs in Seven Days

The fluorescence properties of HSFs labeled with AGQDs (50 μg/mL, 100 μg/mL, 200 μg/mL, 400 μg/mL) for 24 h were detected for seven days, and the detection results were shown in Figure 12 (the picture mainly shows the state of AGQDs-labeled fibroblasts with 400 μg/mL) and Appendix A. The Image J gray scale statistics were performed on the fluorescence intensity of HSFs during these seven days, and the results of cell-fluorescence attenuation were shown in Figure 13. The results showed that after the HSFs were cultured with four concentrations of AGQD for 24 h, after 7 days of fluorescence monitoring of the labeled HSFs, the HSFs labeled with different concentrations of AGQD still showed a different intensity of orange–red fluorescence on the seventh day. From the results of the fluorescence intensity detection of the HSFs, it can be concluded that the fluorescence intensity of 400 μg/mL AGQDs-labeled HSFs was 2–3 times higher than that of 50 μg/mL AGQDs-labeled HSFs. The HSFs labeled with 100 μg/mL and 200 μg/mL AGQDs had the same fluorescence intensity and a basically consistent fluorescence attenuation curve. Moreover, their fluorescence intensity is between 50 μg/mL and 400 μg/mL AGQDs-labeled cells. Compared with 50 μg/mL AGQDs-labeled HSFs, when 100 μg/mL, 200 μg/mL and 400 μg/mL AGQDs were used to label HSFs, their fluorescence intensity decreased significantly in the first three days, and the fluorescence attenuation of 50 μg/mL AGQDs-labeled cells was stable in 7 days. After 7 days of fluorescence attenuation, the fluorescence intensity of cells labeled with 400 μg/mL AGQDs was still stronger than that of 100 μg/mL and 200 μg/mL AGQDs-labeled cells, and the fluorescence intensity of 100 μg/mL and 200 μg/mL AGQDs-labeled cells was basically the same as that of 50 μg/mL AGQDs-labeled cells. The results show that the fluorescence labeling effect of AGQDs on HSFs is long-term. The fluorescence intensity of HSFs labeled with a high concentration of AGQDs was higher than that of HSFs labeled with a low concentration of AGQDs, and the fluorescence intensity attenuation of HSFs labeled with a high concentration of AGQDs was stronger than that of HSFs labeled with a low concentration of AGQDs, and finally tended to be consistent. It is suggested that the binding of AGQDs in cells has lower limit stability in a certain time range.

#### 2.2.6. The Tracer Detection of Cells on Three-Dimensional Scaffolds

The scanning electron microscope picture of the chitosan demethylcellulose sodium scaffold prepared by the same experimental research group is shown in Figure 14. The results of the SEM showed that the three-dimensional scaffolds had a honeycomb structure, which is similar to the geometric structure of the three-dimensional cell scaffold prepared after the decellularization of human skin. Such a honeycomb geometry has the conditions for HSFs to grow on it. The HSFs inoculated on the cell scaffold were followed up for seven days through the fluorescence properties of the cells; the results are shown in Figure 15, Figure 16, Figure 17 and Figure 18. The orange–red part of the dot in the picture is the fibroblast labeled by AGQDs. It can be seen from the figure on the first day, when the distribution density of HSFs on the surface of the scaffold was lower than on the fifth and seventh days, and the distribution depth in the cross-section of the scaffold was less than on the fifth and seventh days. Through the fluorescence properties of HSFs, the distribution of HSFs on the three-dimensional scaffold can be detected, and the purpose of tracking and locating living cells on the three-dimensional scaffold can be achieved.

## 3. Materials and Methods

### 3.1. Reagents and Apparatus

Human normal skin dermal HSFs were donated by the Shanxi Bethune Hospital, Dulbecco’s Modified Eagle Medium (DMEM) high-sugar medium was purchased from Cytiva (American), fetal bovine serum (FBS) was purchased from Gibco (American), CCK-8 kit was purchased from Prell (China), and Graphene Oxide Quantum Dots (GOQDs), Amino Graphene Quantum Dots (AGQDs) and Carboxyl Graphene Quantum Dots (CGQDs) were purchased from Xianfeng Nano (China). Moreover, FRTC-Phallacidin was purchased from Invitrogen (American), 4′,6-diamidino-2-phenylindole (DAPI) was purchased from Solarbio (China), Transwell (24-hole plate, membrane pore diameter 8 μm) was purchased from JET Biofil (China) and F200 Transmission Electron Microscope purchased from Japan. JSM-7100 field mission scanning electron microscope was purchased from Japan, UV-1700 ultraviolet-visible spectrophotometer was purchased from Japan, Thermo Forma CO_2_ incubator was purchased from the United States, Thermo Scientific Microplate Reader was purchased from the United States, Bio Tek Living cell workstations were purchased from the United States and Olympus IX70 Fluorescence Inverted Microscope was purchased from Japan.

### 3.2. Characterization of Graphene Quantum Dots

The GOQDs, AGQDs and CGQDs were examined a byF200 Transmission Electron Microscope and JSM-7100 field emission scanning electron microscope. The ultraviolet-visible absorption spectra of three kinds of Graphene Quantum Dots were measured by a UV-1700 ultraviolet-visible spectrophotometer, and the maximum absorption wavelength was detected.

### 3.3. Biocompatibility of Graphene Quantum Dots

Three kinds of Graphene Quantum Dots were dissolved in DMEM containing 10%FBS, respectively, and were arranged into Graphene Quantum Dots culture medium (GOQDs culture medium, AGQDs culture medium, CGQDs culture medium) with concentration gradients of 0 μg/mL, 50 μg/mL, 100 μg/mL, 200 μg/mL and 400 μg/mL.

#### 3.3.1. Detection of Cytotoxicity of Graphene Quantum Dots

The HSFs were inoculated in 96-well plates and cultured for a period of time until the cells adhered to the wall, and then the HSFs were co-cultured with the GOQDs culture medium with concentrations of 0 μg/mL, 50 μg/mL, 100 μg/mL, 200 μg/mL and 400 μg/mL for 24 h. CCK-8 kit was used to detect the activity of HSFs according to the instructions of the kit [10]. The effects of AGQDs and CGQDs on the activity of HSFs were detected according to the same experimental method.

#### 3.3.2. Detection of the Effect of Graphene Quantum Dots on Cytoskeleton

The HSFs were inoculated into the laser confocal culture dish of 35 mm. After the HSFs were adhered to the wall for a period of time, the HSFs were co-cultured with the GOQDs culture medium with concentrations of 0 μg/mL, 50 μg/mL, 100 μg/mL, 200 μg/mL and 400 μg/mL for 24 h. The HSFs were fixed with 4% paraformaldehyde, permeable by 0.2% TritonX-100, stained with FRTC-Phallacidin, stained with DAPI nucleus [11], and observed by a fluorescence inverted microscope. The effects of AGQDs and CGQDs on the cytoskeleton were detected according to the same experimental method.

#### 3.3.3. Detection of the Effect of Graphene Quantum Dots on Cell Migration

The HSFs were inoculated into the upper chamber of the Transwell chamber, and the HSFs were co-cultured with the HSFs in the GOQDs culture medium of 0 μg/mL, 50 μg/mL, 100 μg/mL, 200 μg/mL and 400 μg/mL for 24 h. The HSFs in the Transwell chamber were fixed with 4% paraformaldehyde, stained with 1% crystal violet [12], and the HSFs on the upper side of the small ependyma were wiped off with cotton swabs. The HSFs on the inferior side of the ependyma of Transwell were photographed with a light microscope. The same experimental method was used to detect the effect of AGQDs and CGQDs on the migration ability of HSFs.

### 3.4. Detection of Fluorescence Properties of Graphene Quantum Dots Cell Labeling

#### 3.4.1. Detection of Fluorescence Labeling Effect of Graphene Quantum Dots on Cells

The HSFs were inoculated in a 6-well plate and cultured for a period of time until the HSFs were adhered to the wall. The HSFs were co-cultured with HSFs in the GOQDs medium with concentrations of 0 μg/mL, 50 μg/mL, 100 μg/mL, 200 μg/mL and 400 μg/mL for 24 h. Replaced with new a culture medium, the fluorescent labeling of HSFs was detected in three bands (blue bands 377 nm–447 nm, green bands 469 nm–525 nm and red bands 531 nm–593 nm) using the living cell workstation. The fluorescence labeling of HSFs by AGQDs and CGQDs was detected according to the same experimental method. Through the above experiments, Graphene Quantum Dots which can label HSFs with fluorescence were selected.

#### 3.4.2. Fluorescence Attenuation Detection of Graphene Quantum Dots Labeled Cells

The Graphene Quantum Dots selected in 3.4.1 were co-cultured with HSFs at the same gradient concentration as in the above experiment for 24 h. The culture medium of Graphene Quantum Dots was removed and replaced with a new 10% FBS DMEM medium without Graphene Quantum Dots. The fluorescence properties of HSFs were detected in the living cell workstation, and the fluorescence detection results of HSFs were taken for the first day. The fluorescence properties of HSFs were detected on the 2nd, 3rd, 4th, 5th, 6th and 7th day, respectively, which was used as the quantitative statistical basis for the fluorescence attenuation of HSFs.

### 3.5. Cell Tracing in Three-Dimensional Scaffolds

The Graphene Quantum Dots that can label HSFs were screened by 3.4.1. Combined with the fluorescence attenuation statistics of 3.4.2, 100 μg/mL Graphene Quantum Dots were used as the final concentration of labeled HSFs. The HSFs were inoculated on a sodium chitosan demethylcellulose scaffold prepared by references. 100 μg/mL Graphene Quantum Dots were added and co-cultured with HSFs for 24 h. The distribution of HSFs in the scaffold was detected by the fluorescence properties of HSFs, and the results were taken as the results of the first day. On the third, fifth and seventh day, the Graphene quantum dot culture medium containing 100 μg/mL was replaced, and the distribution and growth of HSFs in the scaffold was detected. 100 μg/mL Graphene Quantum Dots were co-cultured with cell scaffolds for 1, 3, 5 and 7 days, and no cells were inoculated on the cell scaffolds. The effect of Graphene Quantum Dots on the fluorescence properties of cell scaffolds was observed under the living cell workstation as a blank control group, through the fluorescence properties of cells to achieve the purpose of monitoring cells in a three-dimensional environment for a long time to verify the possibility of using Graphene Quantum Dots to label HSFs on the scaffold.

### 3.6. Statistical Analysis

Graph Pad Prism8 and Image J were used to analyze the data and images. One-way ANOVA was selected, and an LSD test was used for post-comparison. The data were expressed as mean ± standard deviation. The difference was statistically significant (*p* < 0.05).

## 4. Conclusions

There are many ways to label and identify cells in vitro. According to the source of markers, they can be divided into endogenous markers and exogenous markers: (1) Endogenous markers refer to the labeling of cell proteins, genes, etc., which involves the labeling of the target substance in the cell’s genes; its disadvantage is that it may change the cell’s related biological behavior. This makes it difficult for us to study the nature of cells [13,14]. (2) Exogenous markers include targeted and non-targeted dye probes, such as DAPI and Hoechst. [15,16,17,18]. The above cell-labeling methods have different advantages and disadvantages, such as the complex labeling process, short quenching time after living cell labeling and some dyes cannot meet the requirements that cells still have for activity after labeling. Based on the needs of this experiment, it is particularly necessary to find a cell marker with a simple labeling method, long quenching time and no effect on cell biological behavior.

In recent years, quantum dots as fluorescent probes have been widely used in cell imaging [19,20,21,22,23,24]. Among many nano-luminescent materials, Graphene Quantum Dots are a kind of zero-dimensional material derived from the two-dimensional material graphene, which have excellent optical properties and are widely used in cell imaging [9,25,26,27]. For example, the Graphene Quantum Dots were injected into the body of experimental mice, and the whole and some organs of mice were observed after injection for different times. The results show that Graphene Quantum Dots can be distributed in the whole body through the systemic circulatory system of the organism, and the imaging of isolated organs also shows that stable fluorescence signals can be collected in the heart, liver, kidney and other parts in a certain period of time, and it will not cause inflammation and other pathological damage [28]. For example, after HeLa cells were co-cultured with Graphene Quantum Dots for 2 h, graphene quantum labeling around the nucleus of HeLa cells showed obvious fluorescence, and there was no significant decrease in fluorescence intensity in the process of continuous excitation for 10 min. This shows that Graphene Quantum Dots can infiltrate into cells and maintain fluorescence emission, can be effectively absorbed by cells as fluorescent nanoprobes and are a potential substitute for fluorescent dyes [29]. In complex cellular structures, Graphene Quantum Dots can selectively bind/react with bioactive molecules, organelles and ultrastructures in organelles, or respond to the stimulation of the intracellular microenvironment (temperature, pH, polarity, oxygen content and viscosity, etc.) [30,31], resulting in changes in fluorescence properties such as excitation/emission wavelength, fluorescence intensity and fluorescence lifetime. The imaging of cells by a fluorescence probe can be observed by a fluorescence (lifetime) microscope.

With the development of the industry, the production process of Graphene Quantum Dots has become mature, and it is a convenient and effective fluorescent labeling material that can be obtained. The study of cell imaging requires strict requirements for the properties of fluorescent probes such as size, toxicity, biocompatibility, stability, specificity, detectability and fluorescence response [32]. Based on the need for related research on HSFs in three-dimensional scaffolds of tissue engineering, it is key to select Graphene Quantum Dots that can label HSFs, in order to find a convenient way to label fluorescent HSFs for a long time without affecting the cell morphology, cell activity, migration ability and so on.

In this paper, according to the need of labeling living cells in tissue engineering, cytotoxicity and fluorescence labeling experiments were carried out using GOQDs, AGQDs and CGQDs as fluorescent labeling materials. Graphene Quantum Dots (AGQDs), which can make HSFs labeled with fluorescence and do not affect cell activity and migration, were successfully screened. Moreover, the fluorescence attenuation of AGQDs-labeled HSFs was detected, and the possibility that AGQDs could trace HSFs in seven days after one-time labeling for 24 h was obtained. The HSFs were inoculated on a sodium chitosan demethylcellulose scaffold. According to the relative fluorescence attenuation curve of HSFs, a 100 μg/mL AGQDs culture medium was selected as the fibroblast labeling concentration, and the HSFs in the scaffold were labeled. At the same time, the growth and distribution of HSFs in the scaffold was detected by fluorescence on the 1st, 3rd, 5th and 7th day. The results verify the possibility of labeling HSFs with AGQDs and detecting them in three-dimensional scaffolds. In this experiment, we successfully obtained a method that can trace cells for a long time in a three-dimensional environment, and which has the advantages of bearing no toxicity to the tracer cells, no influence on cell migration and has convenient labeling means. It is expected to bring convenience for the study of the biological behavior of HSFs in a three-dimensional environment in tissue engineering.

## Figures and Tables

**Figure 1 ijms-23-11040-f001:**
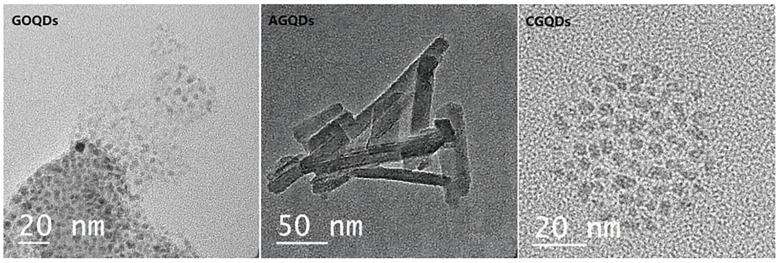
The TEM of three kinds of Graphene Quantum Dots.

**Figure 2 ijms-23-11040-f002:**
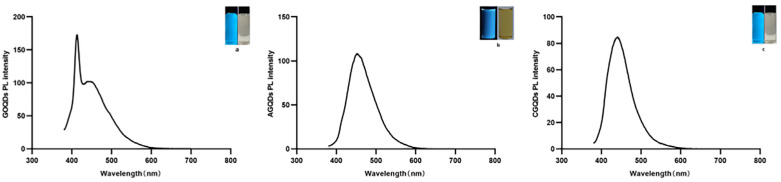
PL spectrum of Graphene Quantum Dots powder dispersed in water. (**a**) Photos of GOQDs solution at 365 nm UV lamp (left) and under natural light (right). (**b**) Photos of AGQDs solution at 365 nm UV lamp (left) and under natural light (right). (**c**) Photos of CQQDs solution at 365 nm UV lamp (left) and under natural light (right).

**Figure 3 ijms-23-11040-f003:**
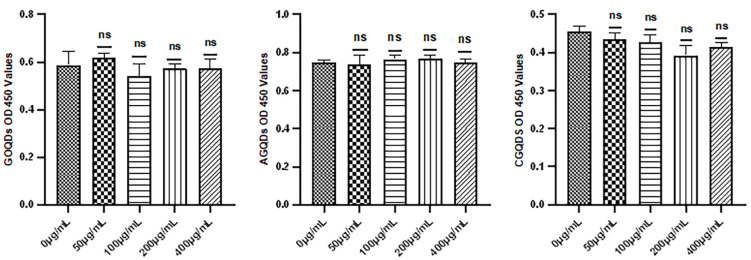
The CCK8 test results of Graphene Quantum Dots. ns: *p* > 0.05.

**Figure 4 ijms-23-11040-f004:**
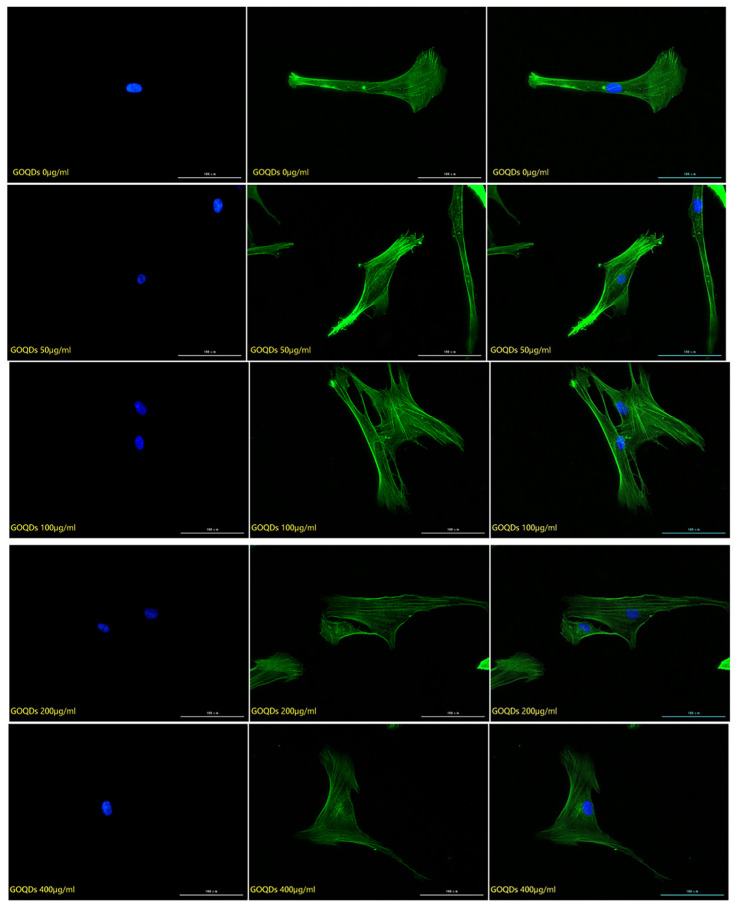
The fluorescence map of FITC-DAPI cytoskeleton staining of HSFs labeled with GOQDs, scale bar: 100 μm.

**Figure 5 ijms-23-11040-f005:**
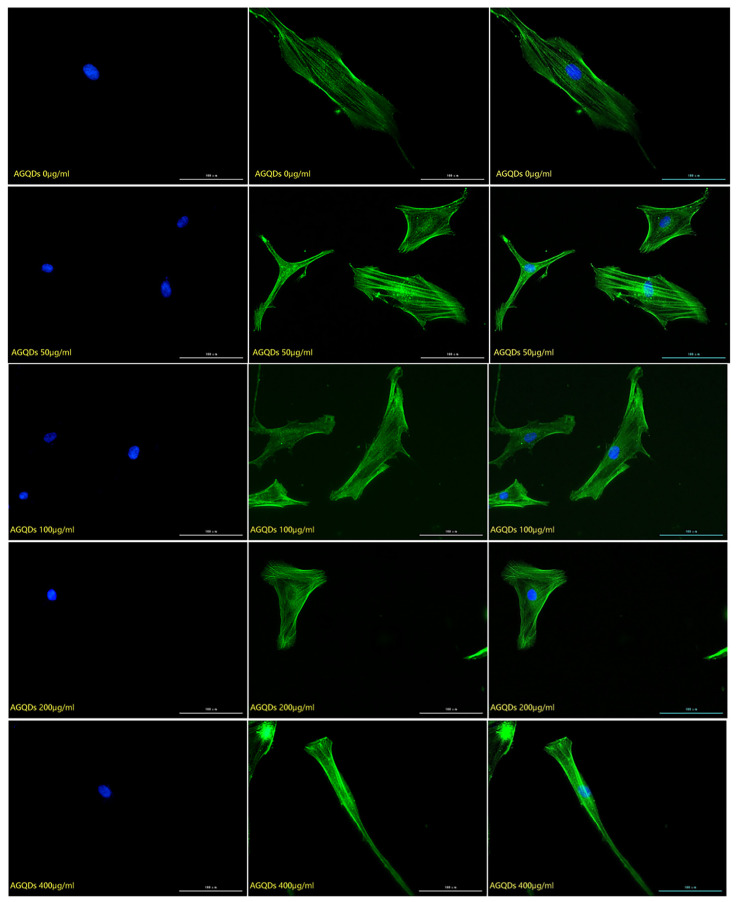
The fluorescence map of FITC-DAPI cytoskeleton staining of HSFs labeled with AGQDs, scale bar: 100 μm.

**Figure 6 ijms-23-11040-f006:**
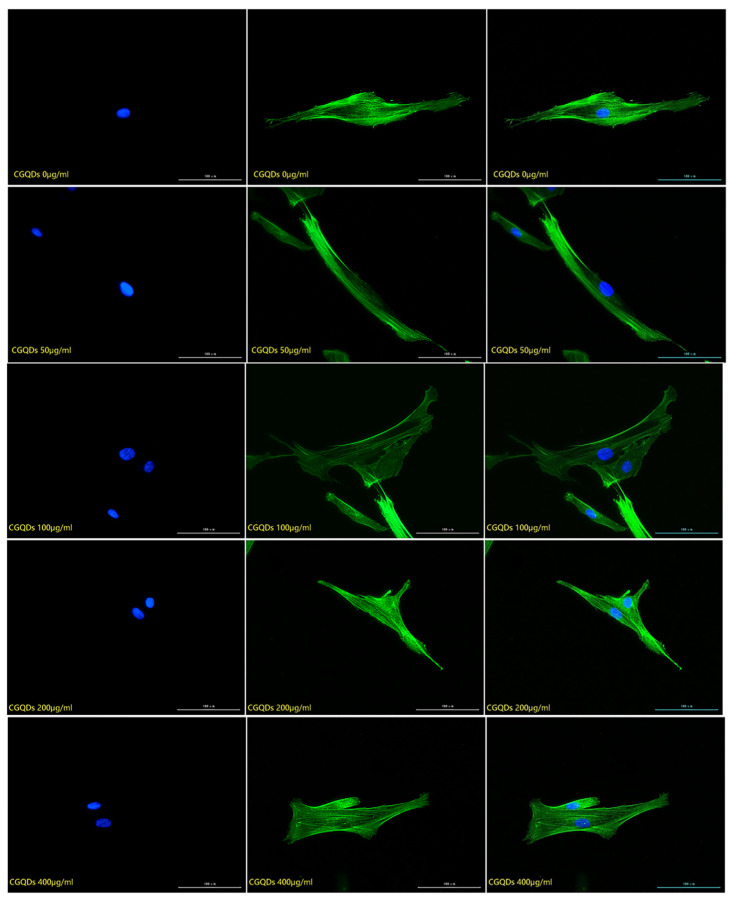
The fluorescence map of FITC-DAPI cytoskeleton staining of HSFs labeled with CGQDs, scale bar: 100 μm.

**Figure 7 ijms-23-11040-f007:**
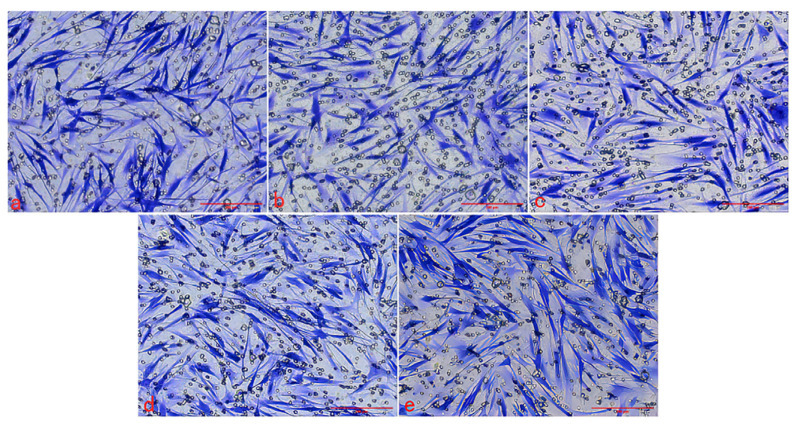
The Transwell cell migration map of HSFs labeled with GOQDs. (**a**–**e**) are 0 μg/mL, 50 μg/mL, 100 μg/mL, 200 μg/mL and 400 μg/mL, respectively. Scale bars: 200 μm.

**Figure 8 ijms-23-11040-f008:**
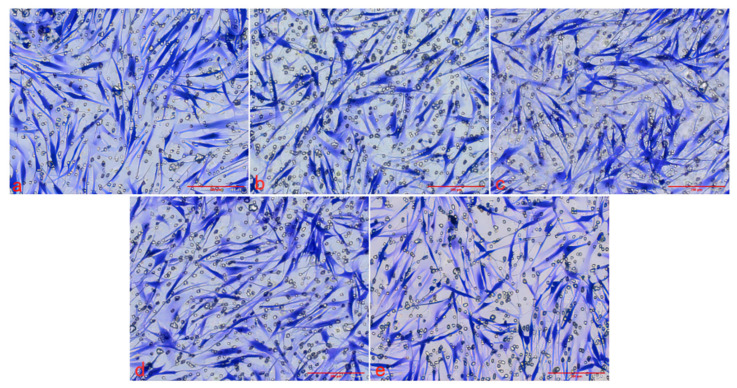
The Transwell cell migration map of HSFs labeled with AGQDs. (**a**–**e**) are 0 μg/mL, 50 μg/mL, 100 μg/mL, 200 μg/mL and 400 μg/mL, respectively. Scale bars: 200 μm.

**Figure 9 ijms-23-11040-f009:**
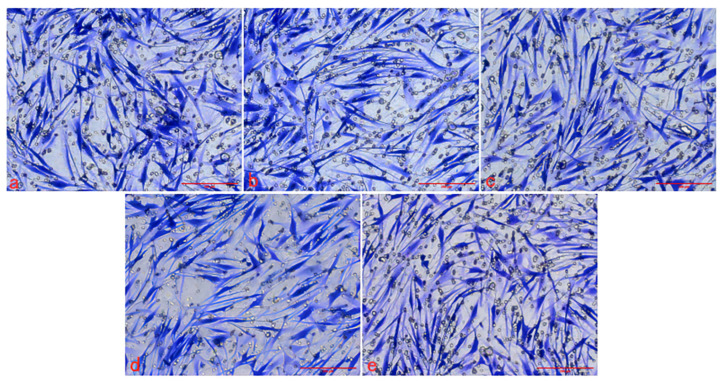
The Transwell cell migration map of HSFs labeled with CGQDs. (**a**–**e**) are 0 μg/mL, 50 μg/mL, 100 μg/mL, 200 μg/mL and 400 μg/mL, respectively. Scale bars: 200 μm.

**Figure 10 ijms-23-11040-f010:**
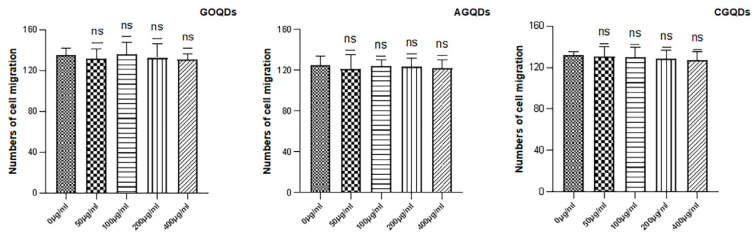
The statistics of the Transwell cell migration map of HSFs labeled with GOQDs, AGQDs and CGQDs, respectively.

**Figure 11 ijms-23-11040-f011:**
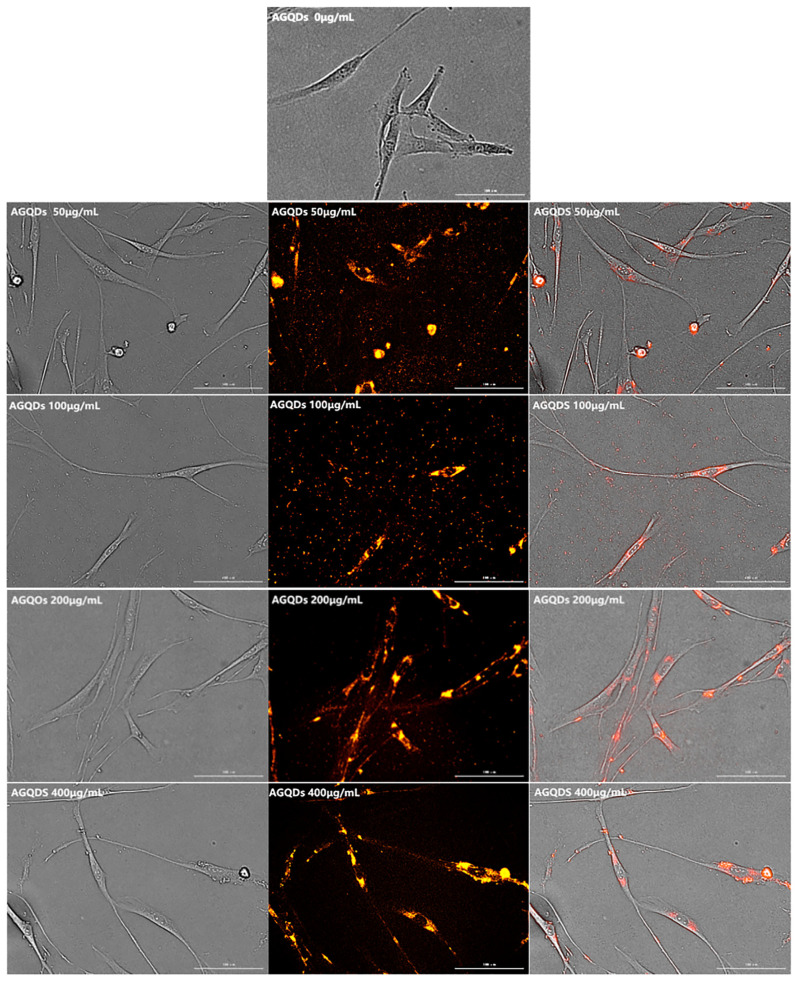
The fluorescence map of HSFs labeled with AGQDs. Scale bars: 100 μm.

**Figure 12 ijms-23-11040-f012:**
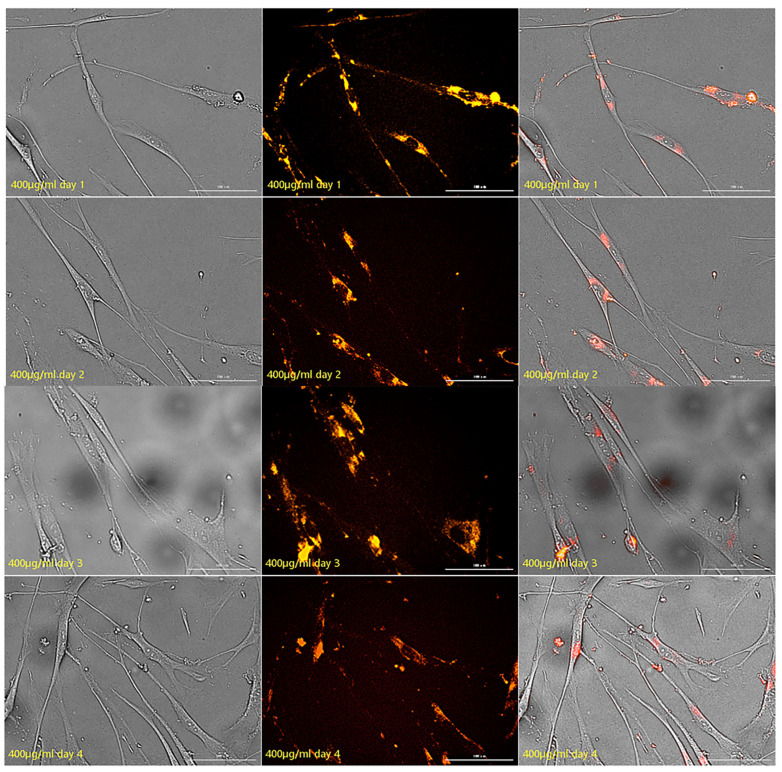
The fluorescence map from the first day to the seventh day, about 400 μg/mL AGQDs-labeled HSFs. From left to right, there are bright field, fluorescent field and composite picture, respectively. Scale bars: 100 μm.

**Figure 13 ijms-23-11040-f013:**
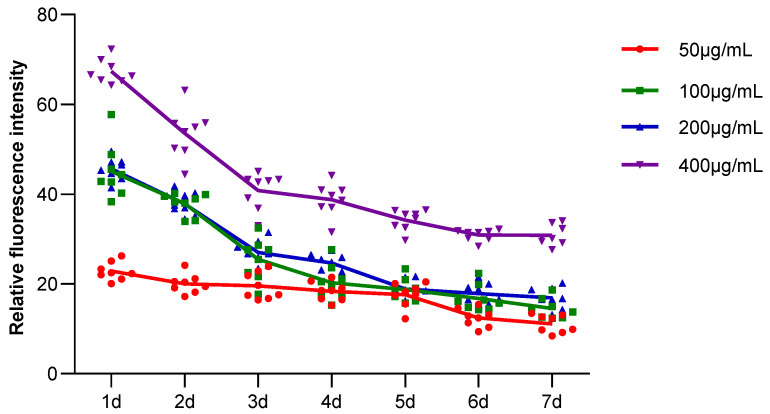
AGQDs-labeled HSFs, 7-day fluorescence attenuation statistics.

**Figure 14 ijms-23-11040-f014:**
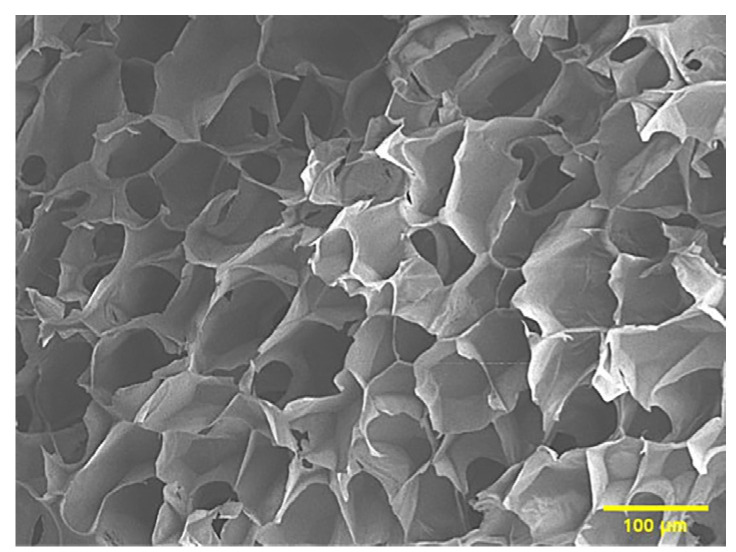
The scanning electron microscope view of chitosan demethylcellulose sodium stent.

**Figure 15 ijms-23-11040-f015:**
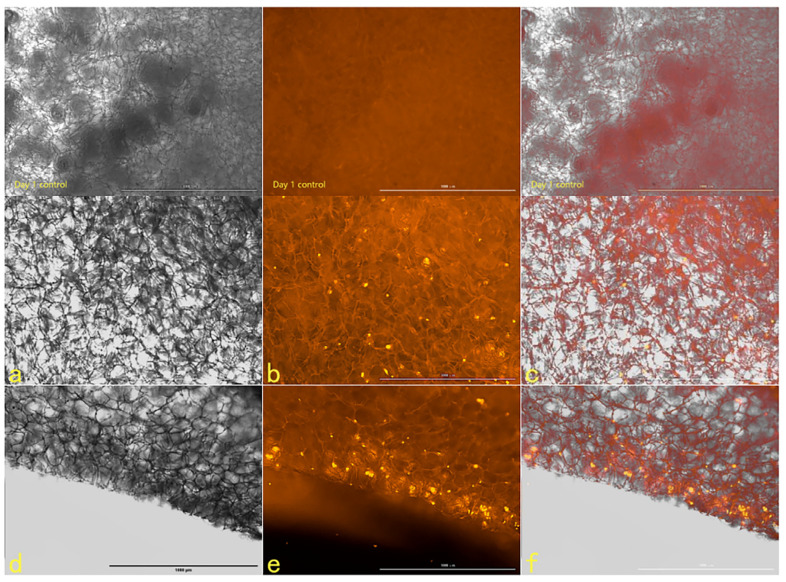
The HSFs detected on a three-dimensional scaffold, day 1. In the picture, (**a**–**c**) are the front of the stent, and (**d**–**f**) are the cross-sections of the stent. (**a**,**d**) are bright fields, (**b**,**e**) are fluorescent fields, and (**c**,**f**) are composite pictures of bright fields and fluorescent fields. Control: only cell scaffolds and a culture medium containing AGQDs, no HSFs. Scale bars: 1000 μm.

**Figure 16 ijms-23-11040-f016:**
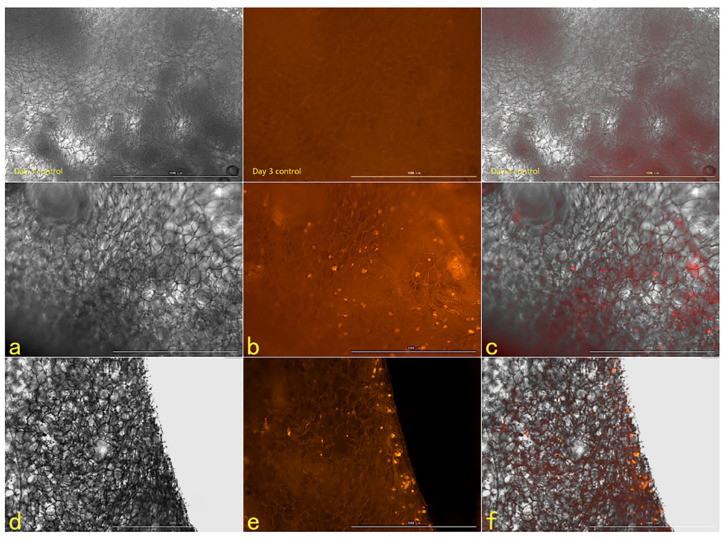
The HSFs detected on a three-dimensional scaffold, day 3. In the picture, (**a**–**c**) are the front of the stent, and (**d**–**f**) are the cross sections of the stent. (**a**,**d**) are bright fields, (**b**,**e**) are fluorescent fields, and (**c**,**f**) are composite pictures of bright fields and fluorescent fields. Control: only cell scaffolds and a culture medium containing AGQDs, no HSFs. Scale bars: 1000 μm.

**Figure 17 ijms-23-11040-f017:**
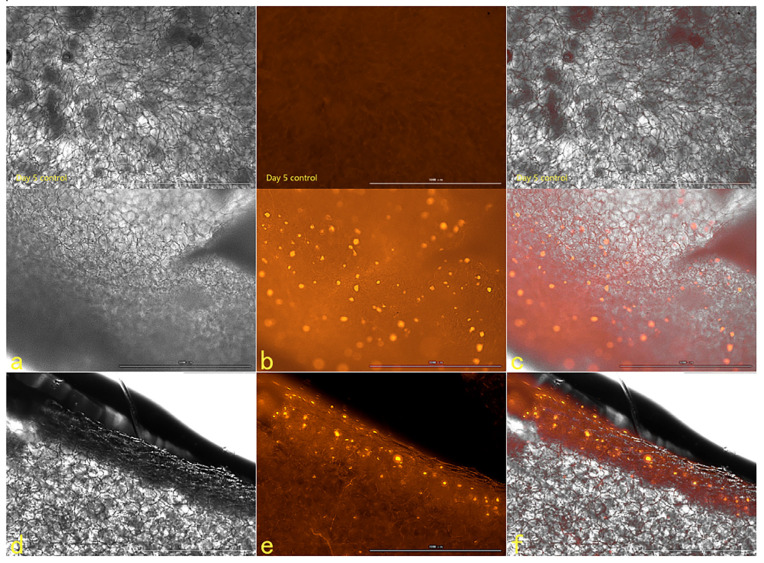
The HSFs detected on a three-dimensional scaffold, day 5. In the picture, (**a**–**c**) are the front of the stent, and (**d**–**f**) are the cross sections of the stent. (**a**,**d**) are bright fields, (**b**,**e**) are fluorescent fields, and (**c**,**f**) are composite pictures of bright fields and fluorescent fields. Control: only cell scaffolds and a culture medium containing AGQDs, no HSFs. Scale bars: 1000 μm.

**Figure 18 ijms-23-11040-f018:**
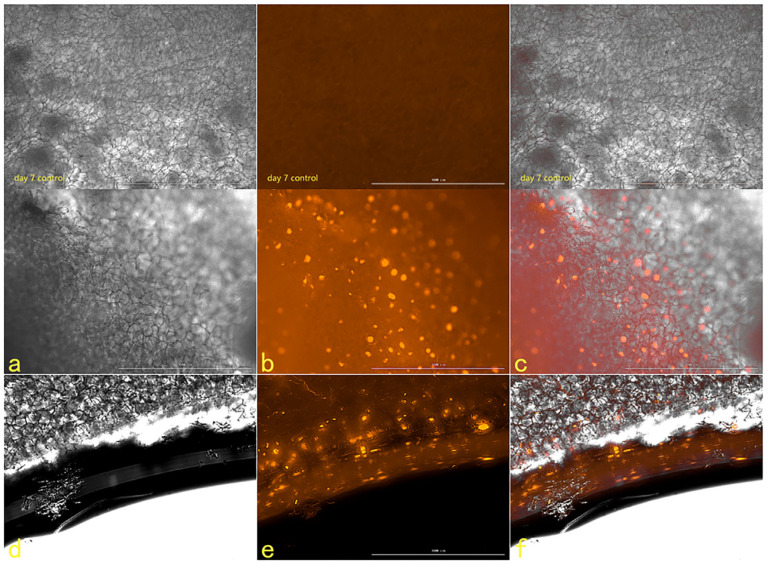
The HSFs detected on a three-dimensional scaffold, day 7. In the picture, (**a**–**c**) are the front of the stent, and (**d**–**f**) are the cross sections of the stent. (**a**,**d**) are bright fields, (**b**,**e**) are fluorescent fields, and (**c**,**f**) are composite pictures of bright fields and fluorescent fields. Control: only cell scaffolds and a culture medium containing AGQDs, no HSFs. Scale bars: 1000 μm.

**Table 1 ijms-23-11040-t001:** The proportion of elements in the three kinds of Graphene Quantum Dots.

Element Proportion (%)	GOQDs	AGQDs	CGQDs
C	47.0	63.1	89.6
N	-	27.0	-
O	38.6	9.8	10.4
Na	14.5	-	-

## Data Availability

Not applicable.

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
