# Peer review of "Study on Long-Term Tracing of Fibroblasts on Three-Dimensional Tissue Engineering Scaffolds Based on Graphene Quantum Dots"

_ijms, 2022, doi:10.3390/ijms231911040_

Round 1
Reviewer 1 Report
The author explains the study and observation regarding their finding, especially the long-term tracing of fibroblast. The presented data are interesting, however, the explanation is too minimal for several figures. Here are several comments to improve the quality of this manuscript.
1. The SEM data in Figure 1. Please delete the all-black shading thing given in the figure (including scale bars), and add the scale bars in the Figures. Because it is quite difficult to see the figure description.
2. Please consider changing the EDS table in Figure 2. It is still stated in Chinese characters or just delete the part because it is already given in Table 1.
3. The description of the axis in Figure 3 should be like, "Wavelength (nm)" instead of "Wavelength/nm", also If the PL doesn't have a unit, it would be better if the author just describes the unit as "a.u".
4. The Figure legend in Figure 4 is not necessary since the x-axis already has the description.
5. Figure 5 is interesting, but the explanation is minimum. Please explain more for clarity. For this figure, we also suggest taking an example from Figure 1 of https://doi.org/10.1016/j.ajpath.2020.01.004.for better figure presentation.
6. Please increase the scale bars of each figure (they are cannot be seen).
7. Please evaluate each figure and please check the visibility of each presented figure description and legend.
8. Actually, the Figure is too many, which makes the explanation not well explained. We suggest the author create supporting information consisting of the data that is not mainly described and explained.
Author Response
1、With regard to the questions raised by the reviewers about the Chinese markings, scale and horizontal and vertical coordinates on the pictures in the article (questions 1-4 and 6), I have modified the pictures in the full text.
2、I have also made a further description and explanation of the problem of cytoskeleton in figure 5 (question 5).
3、I have also made new statistics and analysis on the feasibility of the relevant data in the article.(question 7)
4、For the questions raised by the reviewers about more pictures in the article, they are also put in the following supporting literature(question 8).
Reviewer 2 Report
1. First of all, there are quite a few typing errors in the paper. The author will need to corrected these errors during the revision.
2. Using SEM to observe quantum dots is not a bit unrealistic. If you want to observe the difference between different quantum dots, you should use TEM.
3. As for the results of UV-VIS, the authors need to give a reasonable explanation for the difference of absorption peaks of different quantum dots.
4. Many abbreviations are used before the full name is shown, such as DMEM.
5.section 2.2.2. The authors need to explain the significance of the three pictures expressed by different concentrations of GQDs. I didn't see what the author was trying to say.
As a whole, the authors describe the experimental results, but lack of scientific elaboration and discussion. I don't think the current version is anywhere near ready for publication.
Author Response
1、In response to the question raised by the reviewer that the SEM can not be used to analyze graphene quantum dots, in this article, I replaced the relevant data with graphene quantum dots characterized by TEM(question 2).
2、In response to the question raised by the reviewer that the seedling scanning electron microscope can not be used to analyze graphene quantum dots, in this article, I replaced the relevant data with graphene quantum dots characterized by TEM(question 3).
3、I have also made corresponding corrections to the non-standard use of abbreviations of English words in the article(question 4).
4、The description of cytoskeleton is also analyzed further(question 5).
Round 2
Reviewer 1 Report
Thank you for the change. Before publication, I suggest the author re-check the figure caption again.
1. For supplementary data, please rename the figure as Figure S1, Figure S2, and Figure S3 to avoid misunderstanding with the main Figure.
2. Please change the name in Figure 13 in the supplementary. It should be Figure 3 or Figure S3.
Reviewer 2 Report
The revised version can be accepted.